# Inhibitory Effects of Berberine Hydrochloride on *Trichophyton mentagrophytes* and the Underlying Mechanisms

**DOI:** 10.3390/molecules24040742

**Published:** 2019-02-19

**Authors:** Chen Wen Xiao, Yan Liu, Qiang Wei, Quan An Ji, Ke Li, Li Jun Pan, Guo Lian Bao

**Affiliations:** Institute of Animal Husbandry and Veterinary Science, Zhejiang Academy of Agricultural Sciences, Hangzhou 310021, China; xiaochenwen@zaas.ac.cn (C.W.X.); liuyan@zaas.ac.cn (Y.L.); weiqrabbit@163.com (Q.W.); jiquanan@hotmail.com (Q.A.J.); like@zaas.ac.cn (K.L.); p1198600895@126.com (L.J.P.)

**Keywords:** *T. mentagrophytes*, berberine, underlying mechanisms, inhibitory effects

## Abstract

Background: *T. mentagrophytes* can infect all mammals, including rabbits, causing serious infections with remarkable economic losses for rabbit farmers. Berberine is an alkaloid that is effective against a variety of microbial infections such as *T. mentagrophytes*. Growth curve by dry weight determination and in-vivo antifungal assay were carried out to clarify the inhibitory effect of berberine hydrochloride against *T. mentagrophytes*. Transcriptomics analyses were also carried out for better understanding of the underlying mechanisms. Results: The growth rate of *T. mentagrophytes* was significantly higher in control condition than under berberine hydrochloride or clotrimazole for 60 h. The growth rate of *T. mentagrophytes* was significantly slighter higher in berberine condition (1 mg) than under clotrimazole for 46 h. *T. mentagrophytes* seriously shrunk after berberine or clotrimazole treatment, as observed by TEM and in SEM. Significant recovery was evident in three berberine groups on day 6 compared with the DMSO group. Results from transcriptomics analyses showed 18,881 identified unigenes, including 18,754 and 12,127 in the NT and SwissProt databases. Among these, 12,011, 9174, and 11,679 unigenes belonged to 3 Gene Ontology (GO), 43 KEGG, and 25 KOG categories, respectively. Interestingly, we found that down-regulation of 14α-demethylase exposed to various medicines was slightly different, i.e., berberine hydrochloride (fold change −3.4956) and clotrimazole (fold change −2.1283) caused various degrees of alteration. Conclusions: Berberine hydrochloride could inhibit the growth of *T. mentagrophytes*. Berberine hydrochloride could also cure dermatosis induced by *T. mentagrophytes*. Down-regulation of 14α-demethylase exposed to various medicines was slightly different and might be one of the anti-resistance mechanisms of berberine hydrochloride in *T. mentagrophytes*. The present investigation provides considerable transcript sequence data that would help further assess the antifungal mechanisms against *T. mentagrophytes*, for antifungal medicine development.

## 1. Background 

Dermatophytosis is a common and relevant zoonotic disease of public health concern [1]. It is caused by various species of dermatophytes that damage superficial keratinized tissues in both humans and animals [2,3]. One of the common dermatophyte species in this regard is *Trichophyton mentagrophytes*, which is equally important for humans and animals. Rabbits are frequent hosts of *T. mentagrophytes* [4], which causes serious infections and inflicts substantial economic loss on rabbit farmers [5]. Various antifungal compounds, including clotrimazole, terbinafine, and ketoconazole have been reported for the treatment of dermatophytosis [6]. However, drug resistance, toxicity, and drug-drug interactions limit their application [7]. To identify new classes of antifungal agents to overcome multi-drug resistance mechanisms, many research groups have assessed medicinal plants worldwide [8]. Antimicrobial and antifungal properties of medicinal plant extracts have been widely reported [9]. A previous study has demonstrated that the ethanolic extract of *Phellodendron amurense* has significant antifungal effects both in vitro and in vivo against *T. mentagrophytes* [10]. Berberine hydrochloride constitutes the most important bioactive compound of *Cortex phellodendri*, accounting for approximately 0.6% of the extract from *Berberidis Radix* [11,12]. *Phellodendron amurense* mainly contains about 1.6% *Berberidis Radix*. *Berberidis Radix* is a kind of alkaloid, which is not easy to dissolve in water. It is formed after salt formation with hydrochloric acid. The solubility of berberine hydrochloride is improved and it is easy to dissolve in water. Berberine was shown to inhibit bacteria and other microbes [13,14,15]. Some studies have found that Berberine can inhibit the formation of bacterial extracellular amyloid peptide, thereby interfering with the formation and stability of biofilm [16]. In addition, Berberine can also affect the integrity and permeability of bacterial cell membrane, and bind to some proteins on the membrane, thus affecting the structure and function of proteins. It also affects the expression of bacterial DNA and binds to DNA [17]. High-throughput RNA sequencing (RNA-Seq) constitutes a new potent and affordable method for transcriptomic studies [18]. Indeed, the data produced by this technology include sufficient read coverage for *de novo* transcriptome assembly, gene expression assessment, and gene discovery [19]. Sterol 14α-demethylation as a general part of sterol biosynthetic pathways in eukaryotes has been known and studied for more than 30 years [20]. The enzyme catalyzing this reaction was first purified from Saccharomyces cerevisiae in 1984 [21]. Sterol 14α-demethylase participates in sterol biosynthesis and is an essential requirement for yeast viability. Sterol 14α-demethylase is the target for azole antifungal compounds, and resistance to these drugs and agrochemicals is of significant practical importance [22]. In the present study, berberine hydrochloride displayed antifungal activity against *T. mentagrophytes* dermatophytosis, and Illumina Sequencing for transcriptome analysis was used to explore the mechanism of this antifungal activity.

## 2. Results

### 2.1. Growth Curve by Dry Weight Determination

The growth curves of *T. mentagrophytes* under various concentrations of berberine hydrochloride (0.5, 1, and 2 mg/mL) and clotrimazole (5.0 μg/mL) and DMSO (Dimethylsulfoxide) (5%) during 60 h are shown in Figure 1. At 60 h, *T. mentagrophytes* growth was significantly higher in the control and DMSO group than in berberine hydrochloride and clotrimazole groups. The growth rate of *T. mentagrophytes* was significantly higher in berberine condition (1 mg) than under clotrimazole for 46 h (*p* < 0.05). The growth rate of *T. mentagrophytes* was significantly higher in negative control than under other groups for 58 h (*p* < 0.01). The growth rate of *T. mentagrophytes* was significantly lower in negative condition than under berberine, clotrimazole or DMSO (5%) for 10, 22, 34 h. *T. mentagrophytes* growth was slighter lower than the control for 60 h.

### 2.2. Ultrastructure of T. mentagrophytes after Treatment 

*T. mentagrophytes* significantly shrunk after treatment with berberine hydrochloride or clotrimazole as assessed by transmission electron microscopy (TEM). Interestingly, the cell membrane of *T. mentagrophytes* was destroyed by berberine hydrochloride (1 mg/mL) and clotrimazole (0.5 μg/mL) in scanning electron microscopy (SEM) data. There was no obvious change in the DMSO control group and untreated infected cells (Negative). These data are shown in Figure 2.

### 2.3. In Vivo Antifungal Assay

On day 6, a significant recovery was obtained in the three berberine groups. On day 10, a significant recovery was observed in the clotrimazole group. No significant recovery was found in the DMSO group. These data are shown in Figure 3. Before treatment, all the animals from different groups had symptoms of skin diseases including redness, scaling, exposed patches, or ulceration in challenge places. Animals from berberine groups and clomatrizole group receive a basic cure except for the DMSO group (Figure 4). 

### 2.4. Histological Features of the Skin

PAS (Periodic Acid-Schiff stain) staining was used in this study. Micrographs revealed blue nuclei for skin cells; meanwhile, red dots representing the fungus were mostly found on the skin surface and cuticle (Figure 5). Markedly reduced fungus amounts were obtained in the 4 mg berberine group, and lower amounts of the fungi were found after treatment with 2 mg of berberine or 1 mg of clotrimazole compared with controls; DMSO did not affect fungus amounts. Pictures were taken at pre-treatment and 3 weeks post-treatment from different groups in Figure 5.

### 2.5. Transcriptome Profile of T. mentagrophytes

Total RNA was extracted from *T. mentagrophytes* after different treatments. Equal amounts of RNA from all samples were used to generate a cDNA library for Illumina sequencing, and 452,915,688 reads were obtained. After cleaning for irrelevant reads, 447,742,237 entities were obtained (Table 1). These data were deposited in the Sequence Read Archive of the National Center for Biotechnology Information (NCBI) (accession number, GSE80604; SRA: SRP073785). High-quality reads were assembled into 34,129 transcripts and 20,704 unigenes (Table 2).

18,881 unigenes were identified, including 18,754 and 12,127 in the NT and SwissProt databases. Among these, 12,011, 9175, and 11,679 unigenes belonged to 3 Gene Ontology (GO) (Figure 6a), 43 KEGG (Figure 6e), and 25 KOG categories, respectively (Table 3) (Figure 6d). There were 1823 unigenes not matching known proteins, therefore representing potential new genes. 

In total, 506 differentially expressed genes (DEGs) were detected after treatment with berberine compared to the control group. These are divided into 248 upregulated and 258 downregulated genes (Figure 7). Meanwhile, 757 differentially expressed genes (DEGs) were detected after treatment with clotrimazole compared to the control group. These are divided into 425 upregulated and 332 downregulated genes (Figure 7).

### 2.6. GO, KOG and KEGG Classification

A total of 757 (clotrimazole and control comparison) and 506 (clotrimazole and control comparison) DEGs were mapped to 592 (clotrimazole and control comparison) (Figure 6b) and 1108 (berberine and control comparison) GO entities (*p* < 0.05) (Figure 6c); most GO functions were cellular process, organelle, nucleus, binding, catalytic activity, regulation of biological processes, and organelle. A total of 11,679 unigenes (56.41%) in *T. mentagrophytes* transcriptome were mapped KOG (*p* < 0.05) (Figure 6d). DEGs were mapped to 119 (clotrimazole and control comparison) (Figure 6e) and 151 (berberine and control comparison) (Figure 6f) KEGG categories (*p* < 0.05), most of which were related with Metabolic pathways, Biosynthesis of secondary metabolites, and Microbial metabolism.

### 2.7. Putative Molecular Markers

49,417 SSRs were identified in 14,380 of the 20,704 unigenes (69.45%) using MISA. The Mono-nucleotide SSRs made up the most important fraction (59.76%); tri- and di-nucleotide accounted for 23.16% and 13.75%, respectively (Table 4).

### 2.8. Real-time RT-PCR for Transcriptome Result Validation

Quantitative real-time RT-PCR (qRT-PCR) was used to assess 7 selected genes (TR1318|c0_g1, Accumulation-associated protein; TR5979|c1_g1, Putative fungistatic metabolite; TR10031|c19_g1, Multidrug resistance protein CDR2;TR12215|c1_g1, Cholesterol 7-alpha-monooxygenase; TR3885|c0_g1, Accumulation-associated protein; TR905|c0_g1, Putative ankyrin repeat protein; TR7831|c2_g1, ABC transporter B family member) to validate transcriptome data obtained for *Trichophyton mentagrophytes* treated with berberine chloride or clotrimazole (Figure 8 and Figure 9). qRT-PCR data corroborated Illumina RNA-seq results indicating transcriptome data reflect actual mRNA levels in *Trichophyton mentagrophytes* exposed to different medicines. Using the Pearson correlation comparison, We found berberine compared with clotrimazole (X vs. K)has a high agreement either using illumine (−1) or RT-PCR (−1), then a less agreement occurred with clotrimazole compared with the control using illumine (0.86723) or RT-PCR (0.439577), a least agreement occurred with berberine compared with the control using illumine (0.997609) or RT-PCR (−0.38579) (Table 5).Please see the data from the Appendix A.

## 3. Discussion

According to the U.S. Environmental Protection Agency (EPA), in 1997 approximately 244,000 tons and 37,000 tons of fungicides were sold worldwide and in the United States, respectively [23]. A broad spectrum antifungal activity has been demonstrated for azole fungicides, which are employed both in prevention and cure of fungal infections. Similar to the emerging antibiotic resistance, there is a possibility of developing antifungal resistance as a result of increased use of antifungal products in humans, animals, and plants. Antifungal resistance to azoles, and some of its underlying molecular mechanisms have been reported in agricultural research works [24]. Therefore, identifying new medicinal drugs against fungi with reduced side effects is a necessity.

Despite the reported antifungal properties of several plants, their underlying mechanisms remain largely unknown. Berberine hydrochloride, a major *Cortex Phellodendri* constituent, shows various biological properties [25] including antifungal activity [26,27]. *T. mentagrophytes* is one of the common species of dermatophytes, and is equally important for humans and animals. It is frequently found in rabbits [28] and causes serious infections, with subsequent economic loss to rabbit farmers [5]. In the present study, we found antifungal activity for berberine hydrochloride against *T. mentagrophytes* in animal experiments.

The growing rate of *T. mentagrophytes* treated with untreated control was significantly higher than those obtained in the berberine hydrochloride and clotrimazole groups at 60 h, demonstrating that it was fungicidal at a relatively high berberine hydrochloride dose or clotrimazole dose. Interestingly, we noticed that berberine hydrochloride 0.5 mg is more effective than 1 mg (46 h point), it might be in the early stage of action, low dose treatment may have better effect on fungi. The potential flaws of our present study include lacking an azole resistant *T. mentagrophytes* strain which could be used in vitro inhibition assay using berberine hydrochloride at an inhibitory dose in order to unveil if it is also resistant to berberine hydrochloride at certain concentration. Berberine or clotrimazole altered the shape and cell membrane of *Trichophyton mentagrophytes*, as examined by TEM and SEM, respectively, confirming the antifungal activity of berberine chloride against *T. mentagrophytes*. Similar results were obtained in previously reported studies [10]. Thus, clinical and PAS experiments demonstrated that berberine chloride could cure fungal disease in rabbits effectively.

A comprehensive evaluation of gene expression was carried out with the DEGs method. As shown above, multiple signaling pathways were altered in *T. mentagrophytes* treated with berberine chloride, including metabolic pathways, biosynthesis of secondary metabolites, microbial metabolism. These data were validated by qRT-PCR.

SSRs (simple sequence repeats) represent an informative class of genetic markers (Table 3). They are distributed throughout the coding and non-coding regions of all eukaryotic genomes [29], widely use to characterize genetic diversity, construct linkage maps and even tag genes for the purpose of marker-assisted breeding [30]. SSR markers acquired by our transcriptome analysis are potentially useful for genetic analysis in the *T. mentagrophytes* genome. The genomic variable affecting genetic function may have an evolutionary role. It has reported that the rapid evolution of SSR sequences, specifically for those gains or losses of repeats at certain locus, may provide a molecular basis for adaptation to various environments [31].

The fungal sterol 14α-demethylase is an important ergosterol precursor. Azole compounds exert inhibit yeast and fungal sterol 14α-demethylase, suppressing the synthesis of ergosterol, which is a critical constituent of the membrane in these organisms [32]; blocking the synthesis of ergosterol causes disturbance in cell membrane assembly. Sterol 14α-demethylase belongs to the heme-containing cytochrome P450 (CYP51) superfamily of metabolic proteins. The reaction closely resembles demethylation at C-10 by aromatase [33]; sterol 14α-demethylase oxidatively demethylates C-14 of sterols. In the present study, 14α-demethylase was downregulated upon exposure to either berberine hydrochloride or clotrimazole (Please Table 6). This indicates that both medicines potentially might inhibit the growth of *T. mentagrophytes* by blocking the biosynthesis of ergosterol.

Lathosterol transformation into 7-dehydrocholesterol downstream the squalene pathway of cholesterol biosynthesis involves lathosterol oxidase (5-DES, Δ7-sterol-C5(6)-desaturase), a membrane microsomal enzyme [34], requiring molecular oxygen and remarkably induced by NAD(P)H [35]. This enzyme was proposed to have non-heme iron in its catalytic center, and is inhibited by cyanides, Tiron [36], and various hydrophilic chelators [35], but not affected by carbon monoxide [35]. In the present study, the expression of Lathosterol oxidase in *T. mentagrophytes* was reduced upon exposure to either berberine hydrochloride or clotrimazole, please see Table 6 and Table 7. This suggested that both medicines could inhibit the growth of *T. mentagrophytes* by blocking the biosynthesis of Lathosterol oxidase.

Aldehydes are reduced to related alcohols by members of the aldo–keto reductase (AKR) superfamily, in an NADPH-dependent manner [37]. Detoxification of acrolein occurs through involvement of AKR1B, AKR1B7, AKR1C, AKR7A1, and AKR7A2. AKR1A also participates in metabolic pathways requiring the reduction aldehyde portions in target compounds [38]. Ascorbic acid (AsA) has been reported to be produced by the AKR1A and AKR1B enzymes [39]. We found that the expression of Aflatoxin B1 aldehyde reductase member 2 was significantly increased upon *T. mentagrophytes* exposure to either berberine hydrochloride or clotrimazole (Table 6 and Table 7); this might be the pressure response of *T. mentagrophytes* to degrade berberine hydrochloride or clotrimazole for self-protection against the stress of medicine.

Most eukaryotic cell membranes comprise sterols, which play key roles in sustaining membrane integrity and fluidity. Azoles have been used as sterol biosynthesis inhibitors in systemic antifungal therapy in humans [40]. In the present study, KEGG pathway analysis showed that steroid biosynthesis (map00100) (*p* < 0.0024899) (Figure 10) was very important in the activities clotrimazole against *T. mentagrophytes* while berberine hydrochloride is affection the sterol synthesis. Some of the key genes in this pathway are sterol 14α-demethylase, methylsterol monooxygenase, and sterol 24-C-methyltransferase erg-4. We found that 14α-demethylase, methylsterol monooxygenase and sterol 24-C-methyltransferase erg-4 were down-regulated after treatment with either berberine hydrochloride or clotrimazole. These findings suggested that both medicines could inhibit steroid biosynthesis to prevent the growth of *T. mentagrophytes*. Modification of the cyp51A gene is responsible for resistance of *A. fumigatus* against azoles. Cyp51A counts as one of the two genes encoding sterol C14-demethylase isoforms in the above pathogen. Interestingly, we found that down-regulation of 14α-demethylase exposed to various medicines was slightly different, i.e., berberine hydrochloride (fold change −3.4956) and clotrimazole (fold change −2.1283) (Table 5 and Table 6) caused various degrees of alteration. This might be one of the anti-resistance mechanisms of berberine hydrochloride in *T. mentagrophytes*.

## 4. Methods

### 4.1. Fungi Growth 

Eumycetes from dermopathic rabbits were provided by the Institute of Internal Medicine, Shaoxing District (Shaoxing, China). The Chinese Academy of Medical Sciences confirmed the presence of the *T. mentagrophytes* strain in eumycetes, which was grown in Tryptic Soy Broth (TSB) at 28 °C, TSB contained 17 g of Tryptone, 3 g of Phytone, 5 g of NaCl and 1 liter of distilled water, pH 7.1 [41].

### 4.2. Growth Curve by Dry Weight Determination

Berberine hydrochloride (BC) (>98% purity, Lot No. 20130306) was manufactured by Shanghai Yuanye Biotechnology Co., Ltd. (Shanghai, China). Clotrimazole (CLO) (99% purity, Lot No. 23593-75-1) was provided by BaDaTong Medical Company (Taizhou, Zhejiang, China). Time and concentration effects of berberine and clotrimazole on *T. mentagrophytes* were determined as proposed previously [42]. Briefly, the fungi were cultured at 30,000 cells/mL, in presence of berberine hydrochloride (0.5, 1, or 2 mg/mL) or clotrimazole (5.0 μg/mL) at 37 °C for 0 h, 10 h, 22 h, 34 h, 46 h, 58 h and 60 h, respectively. Duplicate 1 mL aliquots of homogenized samples were removed from each conical flask and transferred to pre-weighed Eppendorf tubes. After centrifugation was performed 13,300× *g* for 20 min, the sediments were dried in an oven at 60 °C. Differential weights were determined using an analytical balance. Each experiment was performed in duplicate. The growth of *T. mentagrophytes* was observed.

### 4.3. Ultra-Structural Analysis by Electron Microscopy 

The fungi were cultured at 30,000 cells/mL, Berberine hydrochloride (1 mg/mL) and Clotrimazole (0.5 μg/mL) were added in fungi for 6 h and the fungi were analyzed as well as DMSO controls. Berberine hydrochloride and clotrimazole were dissolved in DMSO, then dissolved in distilled water, the concentration of DMSO in water was 5%. 

For TEM, the samples were fixed with 2.5% glutaraldehyde in phosphate buffer (0.1 M, pH 7.0; >4 h) followed by post-fixation with 1% OsO_4_ in phosphate buffer (1–2 h). After washing, the specimens were dehydrated by graded ethanol and incubated in acetone. After resin embedding, the samples were sectioned on a LEICA EM UC7 ultratome, followed by staining with uranyl acetate and alkaline lead citrate (each 5 to 10 min) and analysis under a Hitachi Model H-7650 TEM (HITACHI, Tokyo, Japan).

For SEM, the specimens were fixed and post-fixed as described above for TEM. Then, they were transferred into ethanol:iso-amyl acetate (1:1 *v*/*v*) for 30 min, and iso-amyl acetate overnight. After dehydration in a Hitachi Model HCP-2 critical point dryer (HITACHI, Tokyo, Japan) with liquid CO_2_, the specimens were coated with gold-palladium on a Hitachi Model E-1010 ion sputter (4–5 min) before analysis on a Hitachi Model TM-1000 SEM.

### 4.4. Experimental Animals and In Vivo Antifungal Assay

Twenty five New Zealand white male rabbits aged 45 days, weighing 800–950 g, were obtained from the experimental animal center at Zhejiang University, Hangzhou, China. The study was performed after approval from the Bioethics Committee of Zhejiang Academy of Agricultural Sciences (NO.320569), strictly following existing guidelines. The animals were assigned to five groups (*n* = 5). Three groups treated with berberine hydrochloride were classified as Group 1 (4 mg), Group 2 (2 mg), and Group 3 (1 mg). Two additional groups were designated as positive (PC; clotrimazole at 1 mg) and negative (NC; 1 mL DMSO) control groups, respectively.

Dermatophytosis was induced as described in a previous report [43]. Briefly, a 1-mL suspension (1.0 × 10^6^ cells) of *T. mentagrophytes* was used for infection for 3 consecutive days (Day 1, 2, 3). Different doses of berberine (4 mg, 2 mg, 1 mg) were applied with solution form to groups 1, 2, and 3 from Day 4 to Day 6, for 3 consecutive days, they are applied by sprayed. Clotrimazole (1 mg) or DMSO (1 mL) was applied topically in parallel (Table 8). The animals were assessed for symptoms for 22 days, according to a previously described methodology [44]. In this evaluation, the infected area of skin from each rabbit was divided into four equal quadrants and each area was scored as follows: 0, normal; 1, slightly erythematous patches; 2, well-defined redness, swelling, with bristling hairs, bald patches, or scaly areas; 3, large areas of marked redness, scaling, exposed patches, or ulceration in places; 4, partial damage to the covering and loss of hair; and 5, extensive damage to the covering and complete loss of hair. The scores from various treatment groups were compared. The results were compared using a one-way ANOVA and Tukey’s HSD test. A *p* value < 0.05 was considered statistically significant. Photos from each group were taken before treatment and 3 three weeks post-treatment.

### 4.5. Skin Histology PAS Staining

Three skin tissue specimens were randomly obtained at the end of the study from each group, fixed and paraffin embedded, and assessed histologically and after PAS staining [45].

### 4.6. Berberine Hydrochloride Treatment for Transcriptome Analysis

1.0 × 10^7^ CFU in 1 mL normal saline were used in in vitro cultures. BC and clotrimazole treatment groups (*n* = 3) were submitted to 1/2 minimum inhibitory concentrations (MICs) of respective drugs (i.e., 1 mg/mL BC and 0.5 μg/mL Clotrimazole, in DMSO); control animals received DMSO. After 6 h of incubation (28 °C, 150 rpm) and filtration (gauze), the samples were frozen for further analysis.

### 4.7. RNA Purification, cDNA Library Preparation and Illumina Sequencing for Transcriptome Analysis

TRIzol Reagent (Invitrogen) was used for total RNA extraction, as directed by the manufacturer. Then, mRNA was obtained with oligo (dT) magnetic beads and fragmented using fragmentation buffer (70 °C, 4 min). SuperScript II was used to synthesize first strand cDNA; second strand cDNA was obtained after treatment with DNA polymerase I and RNaseH. Double-stranded cDNAs were incubated with T4 DNA polymerase, Klenow Enzyme (NEB), and T4 polynucleotide kinase (NEB). Then, single A base was added with Klenow exo-polymerase; ligation was performed with Adapter by DNA ligase (NEB). Ligation products were submitted to PCR and purified with QIAquick PCR Purification Kit (Qiagen, Dusseldorf, Germany) for the generation of the cDNA library, quantitatively assessed with Qubit (Invitrogen). A cluster of DNA fragments was amplified using bridge PCR on a flow cell chip. After multiple amplification rounds, the products were sequenced on an Illumina HiSeq 4000.

### 4.8. Assembly and Gene Identification

Reads were assembled with the Trinity software (http://trinityrnaseq.sourceforge.net/) as previously described [46]. Contig generation was performed by extension based on overlapping sequence pairs. Then, contigs were joined into transcripts. The longest were considered to be unigenes. The raw transcriptome data have been deposited in the Sequence Read Archive of the National Center for Biotechnology Information (NCBI) (accession No. GSE80604 and SRA: SRP073785).

### 4.9. Functional Annotation for Unigenes

Nt and SwissProt were used for unigene alignment with BLAST (cut-off e-value = 10^−5^). Blast2GO was utilized for Gene ontology (GO) annotation. KOG and KEGG pathway annotations were also carried out with BLAST against Cluster of Orthologous Groups databases and Kyoto Encyclopedia of Genes and Genomes (cut-off e-value = 10^−5^).

### 4.10. Identification of Microsatellites

Unigenes ≥ 1 Kb were submitted to SSR assessment using the Microsatellite Identification tool (MISA; http://pgrc.ipkgatersleben.de/misa/). Detection criteria were perfect repeat motifs of 1–6 bp, with minimum repeat numbers of 10, 6, 5, 5, 5 and 5 for mono-, di-, tri-, tetra-, penta- and hexa-nucleotide microsatellites, respectively.

### 4.11. Identification of Differentially Expressed Genes

Unigene abundance was assessed by mapping the reads from 9 specimens against the reference unigene set with the RSEM software (University of Wisconsin-Madison, Madison, WI, USA) on Bowtie2 [47]. Relative transcript abundance rates were expressed as FPKM values (fragments per kilobase of exon per million mapped reads) [48], derived as R~109C/NL(C is the number of mappable reads falling into the specific unigene; N is the total number of mappable reads; L is the unigene length. Differentially expressed genes were identified by the DEseq software as described previously, with false discovery rate (FDR) correction [49]. Genes were considered to be differentially expressed with Qvalue < 0.05 and |log 2 ratio| > 1.

### 4.12. Real-time PCR 

First strand cDNA was obtained from total RNA (1 µg) with a kit from Promega (Madison, WI, USA) as directed by the manufacturer. Quantitative RT-PCR was performed on an ABI StepOnePlus (Applied Biosystems, Thermo Fisher Scientific, Waltham, MA, USA) with SYBR Green Supermix (TaKaRa) as proposed by the manufacturer. Amplification was performed at 94 °C (10 min), followed by 40 cycles of 94 °C (15 s), 60 °C (31 s), and 72 °C (1 min). Primers are described in Table 9. Cycle threshold (CT) values were obtained, and data analyzed by the 2^−ΔΔCt^ method [50]. Normalization was carried out with 18S, as an endogenous housekeeping gene. Relative mRNA expression levels were provided as mean ± SD from triplicate experiments.

### 4.13. Statistic Analysis

Group comparison was performed by Duncan’s test. GraphPad Prism 5 (GraphPad Software, San Diego, CA, USA) was used for statistical analyses; statistical significance was set at *p* < 0.05. Pearson correlation were used to statistic analysis, determining the agreement between qRT-PCR analysis and Illumina RNA-seq analysis.

## Figures and Tables

**Figure 1 molecules-24-00742-f001:**
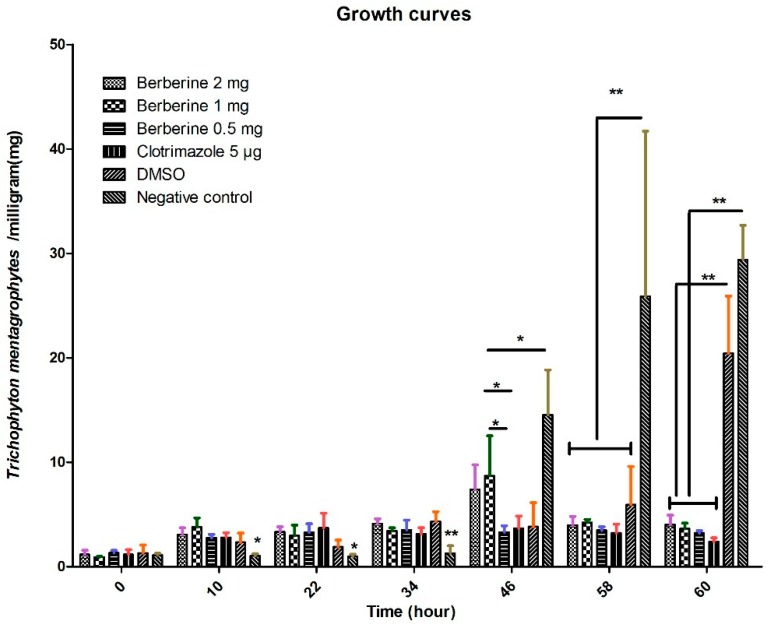
Growth curve by dry weight determination; Growth rates of *T. mentagrophytes* treated with berberine hydrochloride (0.5, 1, and 2 mg/mL) or clotrimazole (5.0 μg/mL). After 60 h, the growth rate of *T. mentagrophytes* treated with control was significantly higher than those of the berberine hydrochloride or clotrimazole groups, but was not significantly different from DMSO treated fungi. * and ** stand for *p* < 0.05 and *p* < 0.01 respectively.

**Figure 2 molecules-24-00742-f002:**
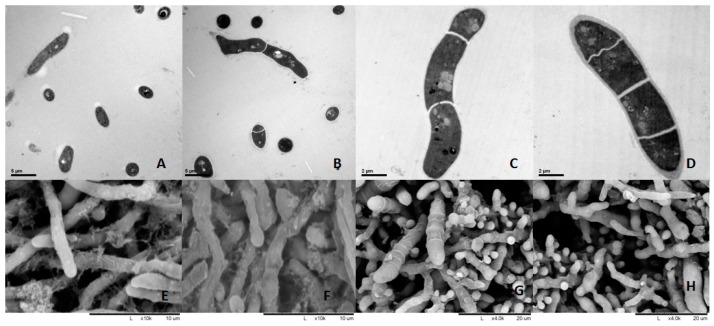
Ultra-structure of fungi submitted to various treatments. (**A**,**E**): 1 mg/mL berberine, (**B**,**F**): 0.5 μg/mL Clotrimazole, (**C**,**G**): Control group was treated with DMSO, (**D**,**H**): Negative group untreated infected cells. (**A**–**D**): Transmission electron microscopy (TEM); (**D**–**G**): Scanning electron microscopy (SEM). *T. mentagrophytes* significantly shrunk after treatment with berberine or clotrimazole as assessed by transmission electron microscopy (TEM) (Figure 2**A**–**D**) and scanning electron microscopy (SEM) (Figure 2**D**–**G**), (**A**) stands for transmission electron microscopy of *T. mentagrophytes* under 1 mg/mL berberine. （**B**） stands for transmission electron microscopy of *T. mentagrophytes* under 0.5 μg/mL Clotrimazole. (**C**) stands for transmission electron microscopy of *T. mentagrophytes* under 0.5 μg/mL DMSO. (**D**) stands for transmission electron microscopy of *T. mentagrophytes* with no treatment. (**E**) stands for scanning electron microscopy of *T. mentagrophytes* under 1 mg/mL berberine. (**F**) stands for transmission electron microscopy of *T. mentagrophytes* under 0.5 μg/mL Clotrimazole. (**G**) stands for transmission electron microscopy of *T. mentagrophytes* under 0.5 μg/mL DMSO. (**H**) stands for transmission electron microscopy of *T. mentagrophytes* with no treatment.

**Figure 3 molecules-24-00742-f003:**
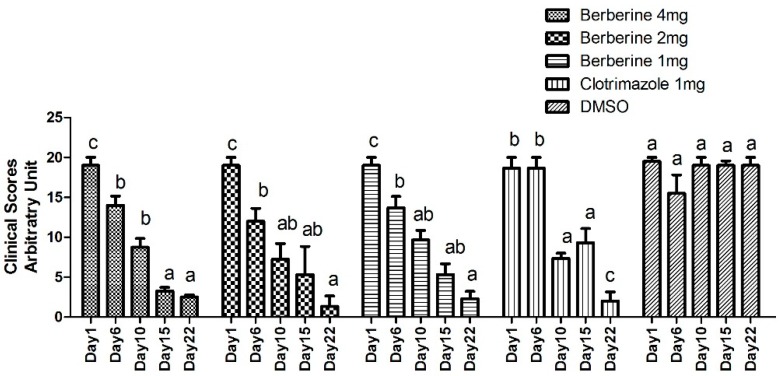
Different doses of berberine were applied to groups 1, 2, and 3 from Day 1, for 3 consecutive days. Clotrimazole or DMSO was applied topically in parallel. The animals were assessed for symptoms for 22 days, according to a previously described methodology In this evaluation, the infected area of skin from each rabbit was divided into four equal quadrants and each area was scored as follows: 0, normal; 1, slightly erythematous patches; 2, well-defined redness, swelling, with bristling hairs, bald patches, or scaly areas; 3, large areas of marked redness, scaling, exposed patches, or ulceration in places; 4, partial damage to the covering and loss of hair; and 5, extensive damage to the covering and complete loss of hair. The scores from various treatment groups were compared. The results were compared using a one-way ANOVA and Tukey’s HSD test. A *p* value < 0.05 was considered statistically significant. a,b,c, stand for significant different from groups, *p* value < 0.05.

**Figure 4 molecules-24-00742-f004:**
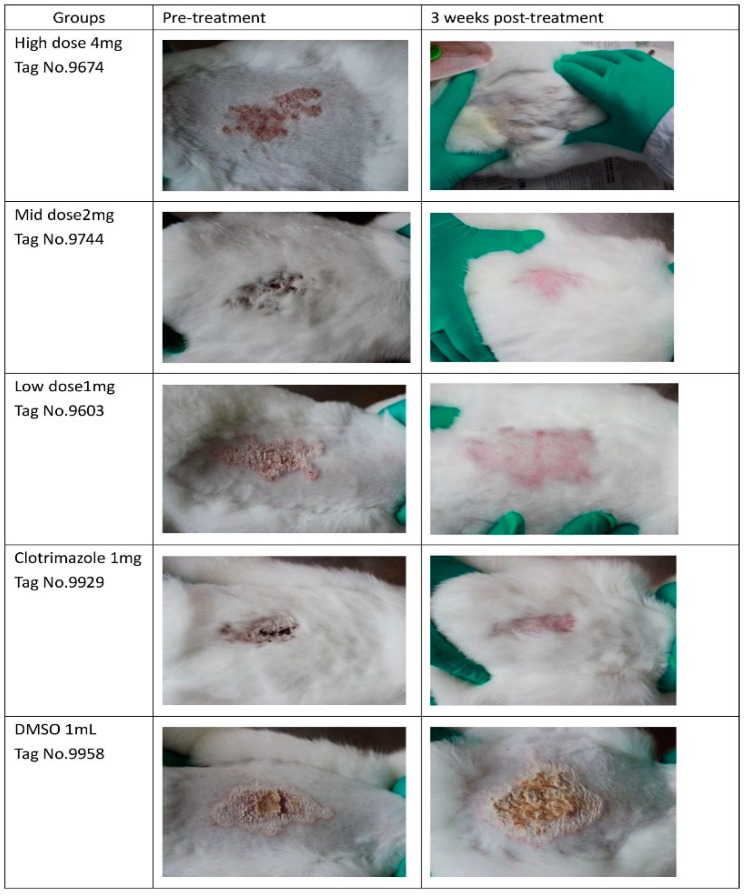
Photos from each group were taken before treatment and three weeks post-treatment.

**Figure 5 molecules-24-00742-f005:**
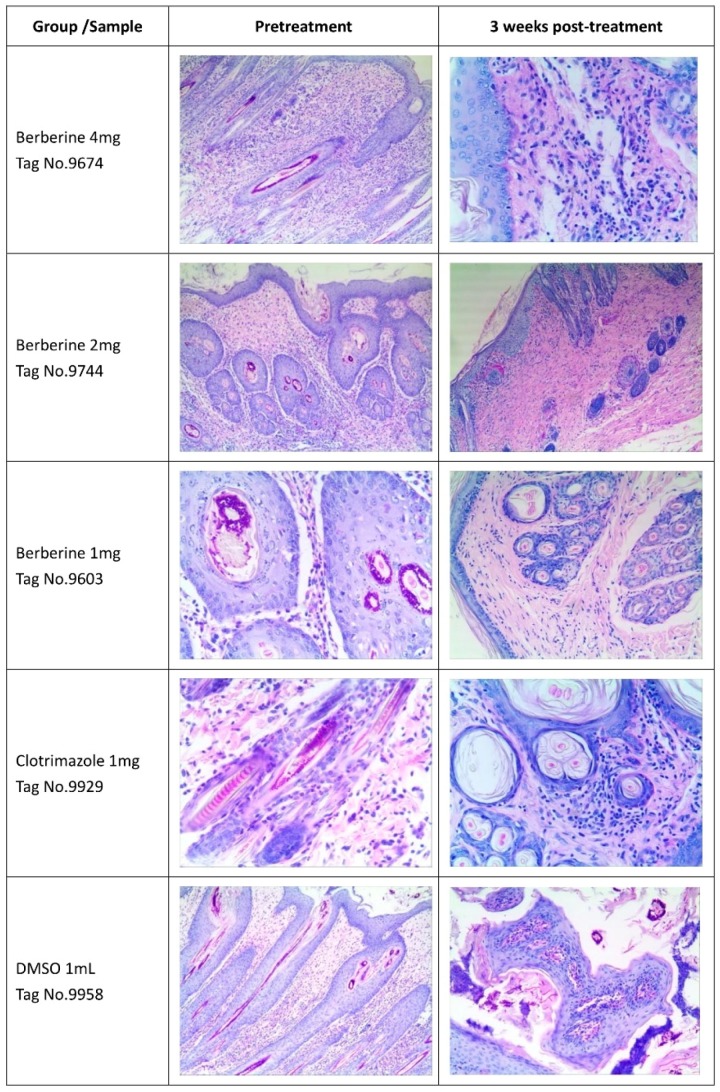
Photographs of skin histology after PAS staining. Red spots represent *Trichophyton mentagrophytes* on the skin. Less fungi were found after treatment with berberine and clotrimazole.

**Figure 6 molecules-24-00742-f006:**
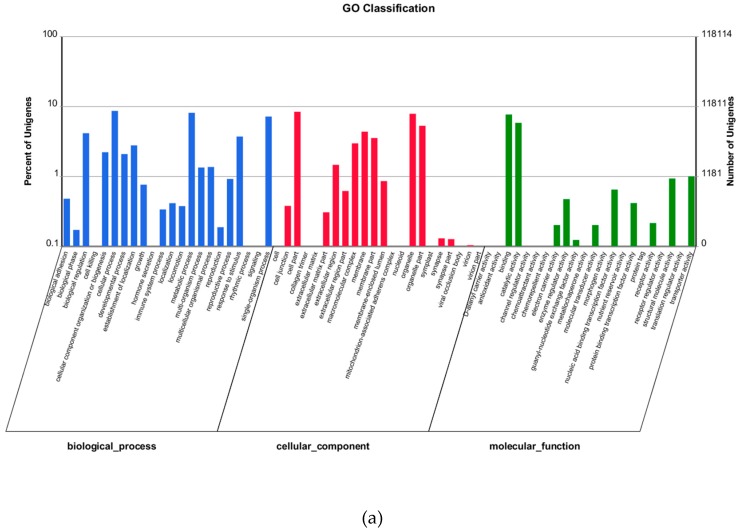
(**a**) Gene ontology (GO) assignment of assembled unigenes of *T. mentagrophytes* Functional annotation of assembled sequences based on gene ontology (GO) categorization. GO analysis was performed by Blast2Go for three main categories: cellular components, molecular function, and biological processes at the second level. The x-axis represents the GO term; the y-axis denotes the number of unigenes. (**b**) Functional annotation of DEGs (clotrimazole and control comparison) based on gene ontology (GO) categorization. GO analysis was performed for three main categories: cellular components, molecular function, and biological processes. The x-axis represents the GO term (the red and green represent all the unigenes and the DEGs annotated to the GO term respectively); the y-axis denotes the number of unigenes. (**c**) Functional annotation of DEGs (berberine and control comparison) based on gene ontology (GO) categorization. The x-axis represents the GO term (the red and green represent all the unigenes and the DEGs annotated to the GO term respectively); the y-axis denotes the number of unigenes. (**d**) Histogram of unigene KOG classification. All the unigene were searched against KOG(euKaryotic Ortholog Groups) databases using the BLASTx algorithm (E-value < 10^−5^). The x-axis indicates the percentage of the number of genes annotation under the group in the total number of genes annotation. The y-axis indicates the groups of KOG. (**e**) KEGG classification of unigenes. The assembled sequences were assigned to the Kyoto Encyclopedia of Genes and Genomes (KEGG) pathways (database downloaded in October 2011) using the online KEGG Automatic Annotation Server (KAAS) (http://www.genome.jp/kegg/kaas/) (BLASTx, cut-off e-value = 10^−5^). (**f**) Statistics of KEGG pathway enrichment. KEGG pathway enrichment were performed R package.

**Figure 7 molecules-24-00742-f007:**
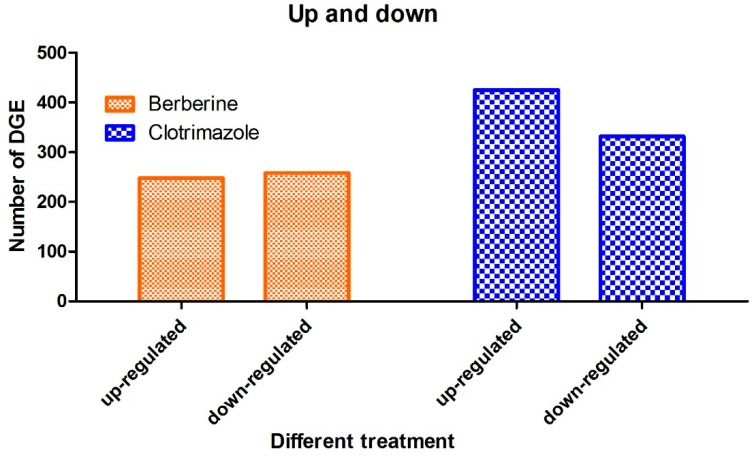
Differentially expressed genes (DEGs) in various groups (*p* ≤ 0.05). A total of 506 DEGs were found in the berberine group, including 248 upregulated and 258 downregulated genes, respectively. Meanwhile, 757 DEGs were identified in the clotrimazole group, including 425 and 332 upregulated and downregulated genes, respectively.

**Figure 8 molecules-24-00742-f008:**
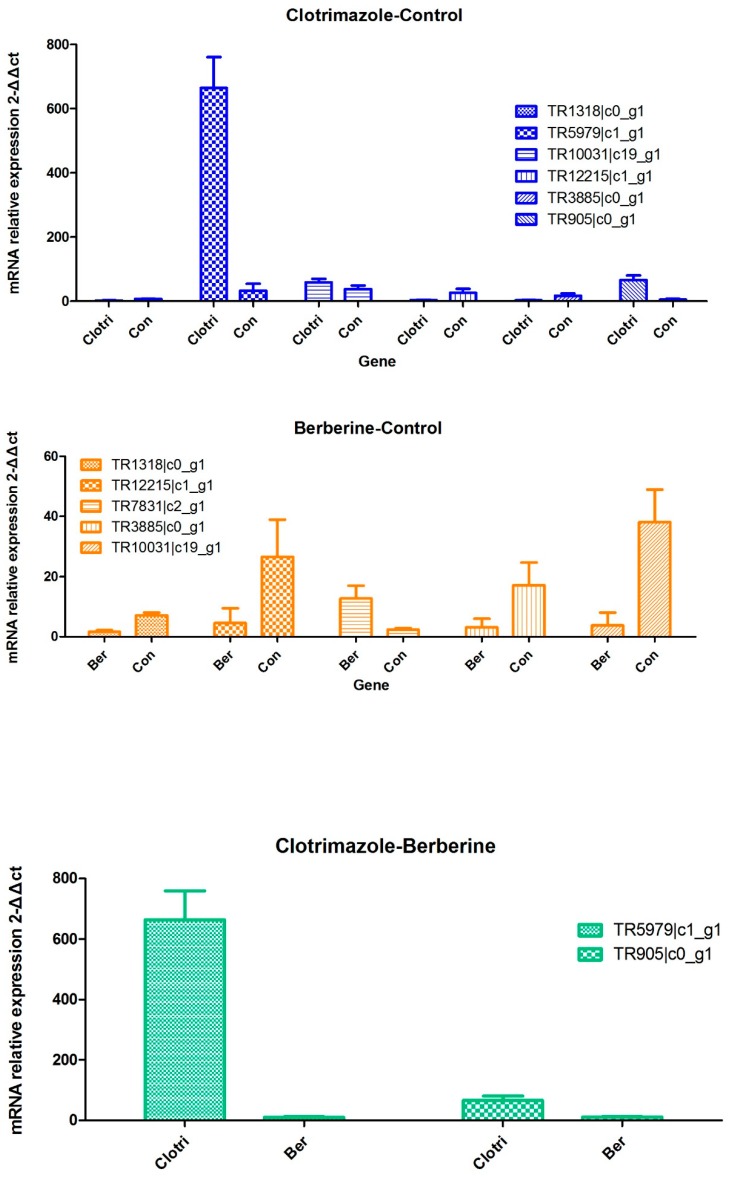
Quantitative real-time RT-PCR (qRT-PCR) was used to assess 7 selected genes (TR1318|c0_g1, Accumulation-associated protein; TR5979|c1_g1, Putative fungistatic metabolite; TR10031|c19_g1, Multidrug resistance protein CDR2; TR12215|c1_g1, Cholesterol 7-alpha-monooxygenase; TR3885|c0_g1, Accumulation-associated protein; TR905|c0_g1, Putative ankyrin repeat protein; TR7831|c2_g1, ABC transporter B family member) to validate transcriptome data obtained for Trichophyton mentagrophytes treated with berberine chloride or clotrimazole.

**Figure 9 molecules-24-00742-f009:**
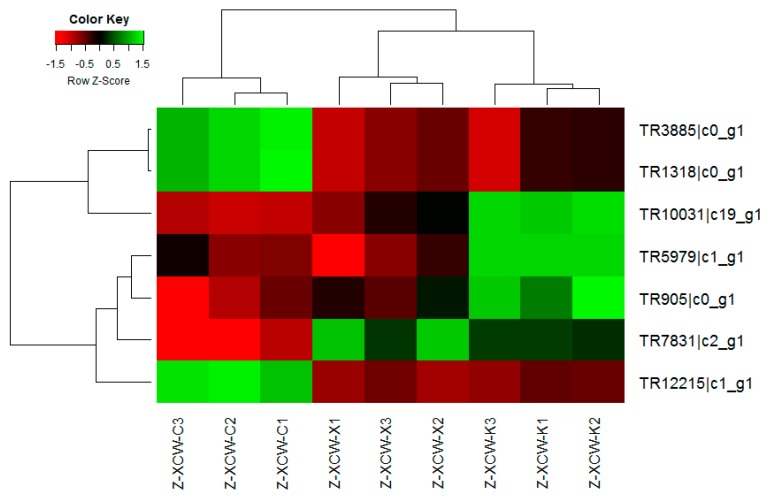
In Figure 9, the heatmap of Illumina RNA-seq data, Z-XCW-X1, Z-XCW-X2, Z-XCW-X3 stands for the berberine group, Z-XCW-K1, Z-XCW-K2, Z-XCW-K3 stands for the clotrimazole group, Z-XCW-C1, Z-XCW-C2, Z-XCW-C3 stands for the control group. Red indicates up-regulation, and green indicates downregulation.

**Figure 10 molecules-24-00742-f010:**
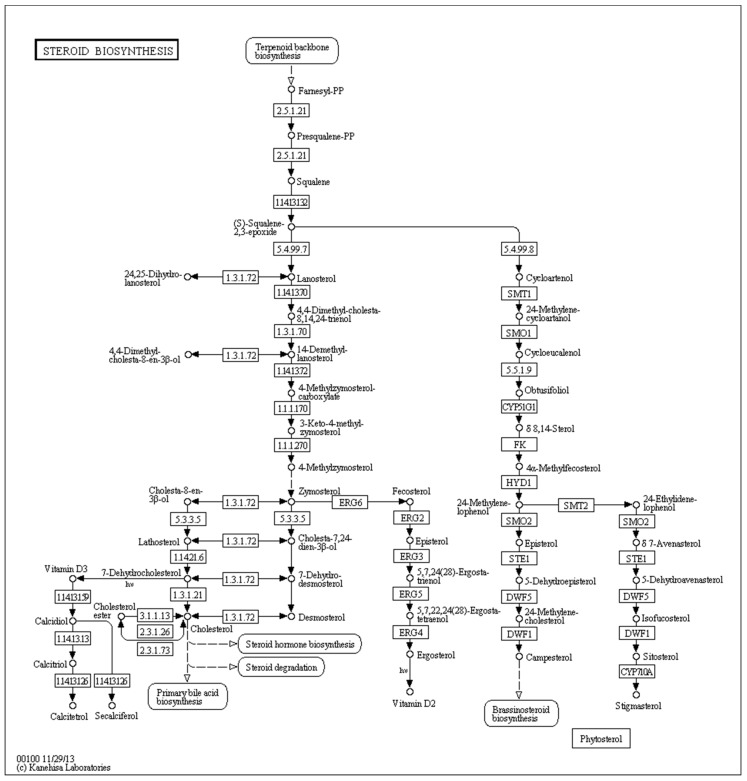
KEGG pathway analysis showed that steroid biosynthesis (map00100) (*p* < 0.0024899) was very important in the activities of berberine hydrochloride or clotrimazole against *T. mentagrophytes*. Some of the key genes in this pathway are sterol 14α-demethylase, methylsterol monooxygenase, and sterol 24-C-methyltransferase erg-4. We found that 14α-demethylase, methylsterol monooxygenase and sterol 24-C-methyltransferase erg-4 were down-regulated after treatment with either berberine hydrochloride or clotrimazole.

**Table 1 molecules-24-00742-t001:** Transcriptome sequence data.

Sample	Direction	Raw Reads	Raw Bases	GC (%)	Q20 (%)	Q30 (%)	Clean Reads	Dropped Reads
Z-XCW-K1	Forward	23776132	2972016500	51	98.83	93.5	23631923	144209
Z-XCW-K1	Reverse	23776132	2972016500	51	98.17	91.56	23322984	453148
Z-XCW-K2	Forward	25648100	3206012500	51	99.03	94.05	25515954	132146
Z-XCW-K2	Reverse	25648100	3206012500	51	98.28	91.84	25210046	438054
Z-XCW-K3	Forward	29748702	3718587750	51	99.09	94.54	29602169	146533
Z-XCW-K3	Reverse	29748702	3718587750	51	98.5	92.81	29303544	445158
Z-XCW-C1	Forward	22619612	2827451500	51	99.05	94.12	22475875	143737
Z-XCW-C1	Reverse	22619612	2827451500	51	98.31	92.11	22261663	357949
Z-XCW-C2	Forward	26527004	3315875500	51	98.98	94.08	26362852	164152
Z-XCW-C2	Reverse	26527004	3315875500	51	98.21	91.85	26056696	470308
Z-XCW-C3	Forward	26040621	3255077625	51	99.15	94.6	25917038	123583
Z-XCW-C3	Reverse	26040621	3255077625	51	98.34	92.13	25610255	430366
Z-XCW-X1	Forward	26266629	3283328625	51	99.13	94.5	26149292	117337
Z-XCW-X1	Reverse	26266629	3283328625	51	98.32	92.05	25829967	436662
Z-XCW-X2	Forward	20853859	2606732375	51	99.1	94.4	20753972	99887
Z-XCW-X2	Reverse	20853859	2606732375	51	98.39	92.28	20520224	333635
Z-XCW-X3	Forward	24977185	3122148125	50	98.71	93.5	24819811	157374
Z-XCW-X3	Reverse	24977185	3122148125	51	97.8	90.46	24397972	579213

**Table 2 molecules-24-00742-t002:** Properties of assembled transcripts and unigenes.

Type	Min Length (bp)	Mean Length (bp)	Median Length (bp)	Max Length (bp)	N50 (bp)	N90 (bp)	Total Nucleotides (bp)	Total Num.
Transcripts	224	3527.83	2490	30991	5845	1881	120401454	34129
Unigenes	224	2605.68	1667	30991	4644	1339	53947990	20704

**Table 3 molecules-24-00742-t003:** Numbers and frequencies of unigenes annotated in public databases.

Search Item	Number of Unigenes	Percentage
Annotated in at least one Database	18881	91.19%
Annotated in GO	12011	58.01%
Annotated in KEGG	9174	44.31%
Annotated in KOG	11679	56.41%
Annotated in NT	18754	90.58%
Annotated in SwissProt	12127	58.57%
Total Unigenes	20704	100.00%

**Table 4 molecules-24-00742-t004:** The information of SSR derived from all unigenes.

Repeat Unit Size	Number of SSRs	Percentage
Monomers	29531	59.76%
Dimers	6793	13.75%
Trimers	11444	23.16%
Tetramers	933	1.89%
Pentamers	221	0.45%
Hexamers	495	1.00%

**Table 5 molecules-24-00742-t005:** Pearson correlation to corroborated with DEGs and Real-time PCR.

	K vs. C	X vs. C	X vs. K
Illumina	0.86723	0.997609	−1
RT-PCR	0.439577	−0.38579	−1

**Table 6 molecules-24-00742-t006:** Select significantly expressed genes of berberine (Z-XCW-X) vs. control (Z-XCW-C).

Gene	Z-XCW-X	Z-XCW-C	log2(Fold Change)	*p*-Value	SwissProt Annotation	KO_annotation
TR12215|c1_g1	2.3085	17.498	−2.9221	7.95 × 10^−10^	Sterol 24-C-methyltransferase erg-4	avermectin B 5-*O*-methyltransferase
TR12246|c0_g1	6.2603	35	−2.4831	8.07 × 10^−6^	Sterol 24-C-methyltransferase erg-4	25/26-hydroxycholesterol 7alpha-hydroxylase
TR13210|c3_g1	31.637	113.53	−1.8434	4.15 × 10^−11^	Lathosterol oxidase	Delta7-sterol 5-desaturase
TR2028|c8_g1	167.25	786.72	−2.2339	1.46 × 10^−5^	Methylsterol monooxygenase	methylsterol monooxygenase
TR2960|c2_g1	33.602	117.54	−1.8066	1.05 × 10^−10^	Lathosterol oxidase	aldehyde decarbonylase
TR3152|c0_g3	0.65461	7.3837	−3.4956	2.83 × 10^−5^	14α-demethylase	epi-isozizaene 5-monooxygenase
TR670|c9_g1	162.3	768.96	−2.2443	1.18 × 10^−5^	Methylsterol monooxygenase	4,4-dimethyl-9beta,19-cyclopropylsterol-4alpha-methyl oxidase
TR7831|c2_g1	55.859	9.4938	2.5567	2.46 × 10^−6^	Multidrug resistance protein 1	ATP-binding cassette, subfamily B (MDR/TAP)
TR4274|c10_g1	127.73	23.609	2.4357	7.78 × 10^−17^	*N*-carbamoyl-l-amino acid hydrolase	allantoate deiminase
TR149|c2_g1	43.157	10.802	1.9983	2.13 × 10^−14^	Leucine-rich repeat extensin-like protein 5	cell wall integrity and stress response component
TR10568|c2_g1	52.393	12.846	2.028	2.53 × 10^−14^	Leucine-rich repeat extensin-like protein 5	cell wall integrity and stress response component
TR11260|c8_g1	118.59	26.708	2.1507	4.8 × 10^−14^	d-amino-acid oxidase	d-aspartate oxidase
TR10117|c2_g1	82.105	19.855	2.048	1.34 × 10^−13^	Aflatoxin B1 aldehyde reductase member 2	aflatoxin B1 aldehyde reductase

**Table 7 molecules-24-00742-t007:** Select significantly expressed genes of clotrimazole (Z-XCW-K) vs. control (Z-XCW-C) in the transcriptome sequence.

Gene	Z-XCW-K	Z-XCW-C	log2(Fold Change)	*p*-Value	SwissProt Annotation	KO_annotation
TR12215|c1_g1	3.0125	17.357	−2.5265	1.83 × 10^−18^	Sterol 24-C-methyltransferase erg-4	avermectin B 5-*O*-methyltransferase
TR12246|c0_g1	6.3583	34.721	−2.4491	2.3 × 10^−15^	Sterol 24-C-methyltransferase erg-4	25/26-hydroxycholesterol 7alpha-hydroxylase
TR13210|c3_g1	69.036	112.67	−0.70671	8.6 × 10^−3^	Lathosterol oxidase	Delta7-sterol 5-desaturase
TR2028|c8_g1	290.97	780.33	−1.4232	1.46 × 10^−3^	Methylsterol monooxygenase	methylsterol monooxygenase
TR2960|c2_g1	71.382	116.64	−0.70848	8.71 × 10^−3^	Lathosterol oxidase	aldehyde decarbonylase
TR3152|c0_g3	1.675	7.3231	−2.1283	1.52 × 10^−3^	14α-demethylase	epi-isozizaene 5-monooxygenase
TR670|c9_g1	285.61	762.72	−1.4171	1.42 × 10^−3^	Methylsterol monooxygenase	4,4-dimethyl-9beta,19-cyclopropylsterol-4alpha-methyl oxidase
TR7831|c2_g1	35.141	9.4307	1.8977	1.08 × 10^−11^	Multidrug resistance protein 1	ATP-binding cassette, subfamily B (MDR/TAP)
TR4274|c10_g1	77.319	23.422	1.723	5.46 × 10^−11^	*N*-carbamoyl-l-amino acid hydrolase	allantoate deiminase
TR149|c2_g1	41.828	10.722	1.9639	3.33 × 10^−14^	Leucine-rich repeat extensin-like protein 5	cell wall integrity and stress response component
TR10568|c2_g1	50.874	12.756	1.9958	9.89 × 10^−15^	Leucine-rich repeat extensin-like protein 5	cell wall integrity and stress response component
TR11260|c8_g1	67.555	26.511	1.3494	1.08 × 10^−7^	d-amino-acid oxidase	d-aspartate oxidase
TR10117|c2_g1	30.774	19.696	0.6438	8.86 × 10^−3^	Aflatoxin B1 aldehyde reductase member 2	aflatoxin B1 aldehyde reductase

**Table 8 molecules-24-00742-t008:** Experimental groups of rabbits.

Group	No. of Animals	Challenge (Day)	Treatment		
			Application	Day	Quantity
Berberine1	5	1, 2, 3	1 mL	4, 5, 6	4 mg
Berberine2	5	1, 2, 3	1 mL	4, 5, 6	2 mg
Berberine3	5	1, 2, 3	1 mL	4, 5, 6	1 mg
PC (Positive control)	5	1, 2, 3	1 mL	4, 5, 6	1 mg
DMSO (Negative control)	5	1, 2, 3	1 mL	4, 5, 6	1 mL

**Table 9 molecules-24-00742-t009:** Real-time PCR primers.

Gene Name	Gene Code	Primer	Sequences (5′–3′)
Accumulation-associated protein	TR1318|c0_g1	F	ACCACCAGCTTCTTATCCAC
		R	AGTCTCCTGCTCACTCGTA
Putative fungistatic metabolite	TR5979|c1_g1	F	CAAATTCATATGCGCCAGT
		R	GCTCGTTCCTCTTATTACCAC
Multidrug resistance protein CDR2	TR10031|c19_g1	F	ATCATCGCCAGTTTGTTCACC
		R	GAGATTTCCACGCTTAACCAC
Cholesterol 7-alpha-monooxygenase	TR12215|c1_g1	F	AATACCCATCCGAAACGCAAG
		R	ACAGCTACTATGACCTCGCAAC
Accumulation-associated protein	TR3885|c0_g1	F	AGAGACATTTGATATCGCTGT
		R	ACCACCAGCTTCTTATCCAC
Putative ankyrin repeat protein	TR905|c0_g1	F	CTAGCTGCTCCTTTCCGTTT
		R	CAACTTTATCTTGACCCGTCCAC
ABC transporter B family member	TR7831|c2_g1	F	ACTGCTTTGCCTCTTATGACC
		R	ATATCTGGCCCCTGAAGTCG
18s rRNA		F	GATACCCGCTGAACTTAAGCA
		R	AATTTGAGCTCTTGCCGCTTC

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
