# Peer review of "Inhibitory Effects of Berberine Hydrochloride on Trichophyton mentagrophytes and the Underlying Mechanisms"

_molecules, 2019, doi:10.3390/molecules24040742_

Round 1

Reviewer 1 Report

Dear authors

the paper presented fore view entitled Inhibitory effects of berberine hydrochloride on TM and the underlying mechanism is sound and the concept of the study seems reasonable in terms of possible clinical implications. The results are nicely presented and the methods are up-to-date. However there are some concerns listed below:

Although in the Figure 1 you present significant reduction of growth following 46 h treatment with Ber you do not mention it in the results nor in discussion. Similar for 58h period. Could you please discuss the notification that Ber 1 mg is more effective than 2 mg (46h point)?

Page 1, line 49. There is nice paper dealing with Ber mechanism (doi: 10.1186/s12906-017-1773-5). Please cite - consider it also in discussing sterol-theory in discussion).

The first sentence of the results does not sound right - revise.

"significantly slighter higher" - please decide is it significant or slighter higher

page 5, line 121/122 - last sentence is obscure

page 11, line 181-185. Please revise the sentences. It is not quite clear what is the clear cut in there results ("a least agreement").

page 12 - there is no figure legend for this figure

page 13 - figure legend i referring to fig 8 is indicated as fig 9

Please provide explanation for differences obtained with Fig 8a.

page 13, line 206 - use corr. English

page 16. lines 269-272 - it is said that steroid biosynthesis was very important in the activities of ber and clo. Actually this is not true since ber is affection sterol synthesis! Please correct.

English language and style should be improved to better present the valuable results!

Author Response

Reviewer 1

the paper presented fore view entitled Inhibitory effects of berberine hydrochloride on TM and the underlying mechanism is sound and the concept of the study seems reasonable in terms of possible clinical implications. The results are nicely presented and the methods are up-to-date. However there are some concerns listed below:

Although in the Figure 1 you present significant reduction of growth following 46 h treatment with Ber you do not mention it in the results nor in discussion. Similar for 58h period. Could you please discuss the notification that Ber 1 mg is more effective than 2 mg (46h point)?

Dear Reviewer, Thank you very much for your comments, we have added the contents in the certain part , please check the section “Growth curve by dry weight determination” and Discussion section.

Page 1, line 49. There is nice paper dealing with Ber mechanism (doi: 10.1186/s12906-017-1773-5). Please cite - consider it also in discussing sterol-theory in discussion).

Dear Reviewer, Thank you very much for your comments, we have added the reference , please check. Thanks so much.

The first sentence of the results does not sound right - revise.

Dear Reviewer, Thank you very much for your comments, we have revise the sentence, please check the first sentence of the results.

"significantly slighter higher" - please decide is it significant or slighter higher

Dear Reviewer, Thank you very much for your comments, we have revise the sentence as "significantly higher" , please check.

page 5, line 121/122 - last sentence is obscure

Dear Reviewer, Thank you very much for your comments, we have revise the sentence, please check.” Results section-Histological features of the skin”

page 11, line 181-185. Please revise the sentences. It is not quite clear what is the clear cut in there results ("a least agreement").

Dear Reviewer, Thank you very much for your comments, we have revise the sentence, please check. “Results section-Real-time RT-PCR for transcriptome result validation”

page 12 - there is no figure legend for this figure

Dear Reviewer, Thank you very much for your comments, we have added the figure legend, please check. Thanks so much.

page 13 - figure legend i referring to fig 8 is indicated as fig 9

Dear Reviewer, Thank you very much for your comments, we have revised it , please check.

Please provide explanation for differences obtained with Fig 8a.

Dear Reviewer, Thank you very much for your comments, Quantitative real-time RT-PCR (qRT-PCR) was used to assess 7 selected genes. But in different comparison, we select different number of genes because we only want to verify the result of Illumina RNA-seq results.

page 13, line 206 - use corr. English

Dear Reviewer, Thank you very much for your comments, we have revise it , please check it in Discussion-1st paragraph.

page 16. lines 269-272 - it is said that steroid biosynthesis was very important in the activities of ber and clo. Actually this is not true since ber is affection sterol synthesis! Please correct.

Dear Reviewer, Thank you very much for your comments, we have revise it , please check it in Discussion- last paragraph.

English language and style should be improved to better present the valuable results!

Dear Reviewer, Thank you very much for your comments, actually, our paper has been edited by a English language Editing Company, Medsci company which is located in Shanghai, China. Thanks so much.

Reviewer 2 Report

The paper in general is well written and explores a new antifungal compound against T. mentagrophytes, a pathogen for mammalians. The paper is valuable given that a double approach has been performed, both in vitro and in vivo assays were performed in a successful way. Additionally, an RNA-seq analysis has been carried out in order to unveil the mechanism of action of two antifungal compounds. This experiment seems to be also correctly validated with some gene expression measurement by qRT-PCR. However, my main criticism is about the way of overcoming the fungal resistance through inhibiting the sterol synthesis. “Most eukaryotic cell membranes comprise sterols, which play key roles in sustaining membrane integrity and fluidity. Azoles have been used as sterol biosynthesis inhibitors in systemic antifungal therapy in humans [37]. In the present study, KEGG pathway analysis showed that steroid biosynthesis (map00100) (P<0.0024899)(Figure 9) was very important in the activities of berberine hydrochloride or clotrimazole against T. mentagrophytes. Some of the key genes in this pathway are sterol 14α-demethylase, methylsterol monooxygenase, and sterol 24-C-methyltransferase erg-4. We found that 14α-demethylase, methylsterol monooxygenase and sterol 24-C-methyltransferase erg-4 were down-regulated after treatment with either berberine hydrochloride or clotrimazole.”

If it is asserted that a problem with the antifungal resistance is growing, it seems to be contradictory to find a compound with a shared mechanism of action that the azole compounds. I would suggest to perform at least an in vitro inhibition assay using berberine hydrochloride at a inhibitory dose released by the present study with an azole resistant T. mentagrophytes strain in order to unveil if also it is resistant to this compound.

Other comments:

Background is not clear. No links between different ideas. several plants are presented without explaining with more detail.

Line 45: Capital letter for phellodendron amurense

lines46-48 are unclear. Link between Berberinee hydrochloride, Cortex phellodendri and Berberidis Radix it is not clear. By the way, Berberidis Radix should be in italic?

Objective is missing

Line 139: genes(Figure7) spaces missing.

P, from p-values sometimes appears in capital letter, and never in italic.

Line 215: I would suggest to say untreated control instead of control.

Lines 215-219: I would be cautious to say the word kill, it depends on the dose, so I would say fungistatic or may be fungicidal at relativeley high berberine hydrochloride dose.

Line 232: Space between role.It

Line 233: comma before specifically

Line 276: A. fumigatus

Author Response

Reviewer 2

The paper in general is well written and explores a new antifungal compound against T. mentagrophytes, a pathogen for mammalians. The paper is valuable given that a double approach has been performed, both in vitro and in vivo assays were performed in a successful way. Additionally, an RNA-seq analysis has been carried out in order to unveil the mechanism of action of two antifungal compounds. This experiment seems to be also correctly validated with some gene expression measurement by qRT-PCR. However, my main criticism is about the way of overcoming the fungal resistance through inhibiting the sterol synthesis. “Most eukaryotic cell membranes comprise sterols, which play key roles in sustaining membrane integrity and fluidity. Azoles have been used as sterol biosynthesis inhibitors in systemic antifungal therapy in humans [37]. In the present study, KEGG pathway analysis showed that steroid biosynthesis (map00100) (P<0.0024899)(Figure 9) was very important in the activities of berberine hydrochloride or clotrimazole against T. mentagrophytes. Some of the key genes in this pathway are sterol 14α-demethylase, methylsterol monooxygenase, and sterol 24-C-methyltransferase erg-4. We found that 14α-demethylase, methylsterol monooxygenase and sterol 24-C-methyltransferase erg-4 were down-regulated after treatment with either berberine hydrochloride or clotrimazole.”

If it is asserted that a problem with the antifungal resistance is growing, it seems to be contradictory to find a compound with a shared mechanism of action that the azole compounds. I would suggest to perform at least an in vitro inhibition assay using berberine hydrochloride at a inhibitory dose released by the present study with an azole resistant T. mentagrophytes strain in order to unveil if also it is resistant to this compound.

Dear Reviewer, Thank you very much for your comments, however we didn’t have an azole resistant T. mentagrophytes strain at present time, we will do this experiment as soon as we get this kind of strain, thank you very much. We have revise the part , please check. please check it in Discussion- last paragraph.

Other comments:

Background is not clear. No links between different ideas. several plants are presented without explaining with more detail.

Dear Reviewer, Thank you very much for your comments, we have revise the part , please check in Background.

Line 45: Capital letter for phellodendron amurense

Dear Reviewer, Thank you very much for your comments, we have revise the part , please check.

lines46-48 are unclear. Link between Berberinee hydrochloride, Cortex phellodendri and Berberidis Radix it is not clear. By the way, Berberidis Radix should be in italic?

Dear Reviewer, Thank you very much for your comments, we have revise the part , please check.

Phellodendron amurense mainly contains about 1.6% Berberidis Radix. Berberidis Radix is a kind of alkaloid, which is not easy to dissolve in water. It is formed after salt formation with hydrochloric acid. The solubility of berberine hydrochloride is improved and it is easy to dissolve in water. Yes, Berberidis Radix should be in italic.

Objective is missing

Dear Reviewer, Thank you very much for your comments, however, we didn’t find the place of this mistake, please inform us, thank you very much.

Line 139: genes(Figure7) spaces missing.

Dear Reviewer, Thank you very much for your comments, we have revise the part, please check.

P, from p-values sometimes appears in capital letter, and never in italic.

Dear Reviewer, Thank you very much for your comments, we have revise in certain parts, please check.

Line 215: I would suggest to say untreated control instead of control.

Dear Reviewer, Thank you very much for your comments, we have revise the part , please check.

Lines 215-219: I would be cautious to say the word kill, it depends on the dose, so I would say fungistatic or may be fungicidal at relativeley high berberine hydrochloride dose.

Dear Reviewer, Thank you very much for your comments, we have revise the part, please check in section “Discussion -3rd paragraph”. Thank you very much.

Line 232: Space between role.It

Dear Reviewer, Thank you very much for your comments, we have revise the part, please check.

Line 233: comma before specifically

Dear Reviewer, Thank you very much for your comments, we have revise the part, please check.

Line 276: A. fumigatus

Dear Reviewer, Thank you very much for your comments, we have revise the part, please check. Thank you very much.

Round 2

Reviewer 2 Report

The paper has been amended in generall, however the main assay missing is not provided in the current version. I strongly recommend to perform it before publication as I explained in the last version. Given the raising fungal resistances it should be easy to find an azole-resistant strain to buy it.

Author Response

Dear Reviewer, Thank you very much for your comments, we have tried to find an azole resistant T. mentagrophytes strain, but we couldn’t find it through certain places such as China General Microbiological Culture Collection Center, CGMCC), and only the normal strain (no: BNCC340405) could be found through website of www.bnbio.com and strain (no:NBRC6202) through website of www.biofeng.com/junzhu/biaozhunjunzhu/NBRC6202.html; We will do this experiment when we get this kind of strain from the clinical field or the other places, thank you very much again. However, we have added a critical discussion to discuss the potential flaws of our study in the “Discussion- third paragraph”. Please check it. Thanks so much.
